# Wheat Teacher: A One-Stage Anchor-Based Semi-Supervised Wheat Head Detector Utilizing Pseudo-Labeling and Consistency Regularization Methods

Rui Zhang [ID], Mingwei Yao [ID], Zijie Qiu [ID], Lizhuo Zhang, Wei Li and Yue Shen *

College of Information and Intelligence, Hunan Agricultural University, Changsha 410125, China;
* Correspondence: shenyue@hunau.edu.cn

**Abstract:** Wheat breeding heavily relies on the observation of various traits during the wheat growth process. Among all traits, wheat head density stands out as a particularly crucial characteristic. Despite the realization of high-throughput phenotypic data collection for wheat, the development of efficient and robust models for extracting traits from raw data remains a significant challenge. Numerous fully supervised target detection algorithms have been employed to address the wheat head detection problem. However, constrained by the exorbitant cost of dataset creation, especially the manual annotation cost, fully supervised target detection algorithms struggle to unleash their full potential. Semi-supervised training methods can leverage unlabeled data to enhance model performance, addressing the issue of insufficient labeled data. This paper introduces a one-stage anchor-based semi-supervised wheat head detector, named "Wheat Teacher", which combines two semi-supervised methods, pseudo-labeling, and consistency regularization. Furthermore, two novel dynamic threshold components, Pseudo-label Dynamic Allocator and Loss Dynamic Threshold, are designed specifically for wheat head detection scenarios to allocate pseudo-labels and filter losses. We conducted detailed experiments on the largest wheat head public dataset, GWHD2021. Compared with various types of detectors, Wheat Teacher achieved a mAP0.5 of 92.8% with only 20% labeled data. This result surpassed the test outcomes of two fully supervised object detection models trained with 100% labeled data, and the difference with the other two fully supervised models trained with 100% labeled data was within 1%. Moreover, Wheat Teacher exhibits improvements of 2.1%, 3.6%, 5.1%, 37.7%, and 25.8% in mAP0.5 under different labeled data usage ratios of 20%, 10%, 5%, 2%, and 1%, respectively, validating the effectiveness of our semi-supervised approach. These experiments demonstrate the significant potential of Wheat Teacher in wheat head detection.

**Keywords:** digital agriculture; deep learning; semi-supervised object detection; wheat head detection

## 1. Introduction

Wheat has consistently been one of the primary crops for human sustenance. Scientists in wheat breeding have persistently strived to develop new wheat varieties with higher yields. Traditional breeding methods still heavily rely on manual observation, and innovative gains in genetic improvement may arise from genomic selection, novel high-throughput phenotyping technologies, or a combination of both [1–4]. These techniques are crucial for selecting important wheat traits related to yield potential, disease resistance, or adaptation to abiotic stress. Despite the realization of high-throughput phenotypic data collection, the development of efficient and robust models for extracting traits from raw data remains a significant challenge. Among all traits, wheat head density (the number of wheat heads per unit ground area) is a major yield component and is still manually evaluated in breeding trials. This manual evaluation is labor-intensive and introduces measurement errors of approximately 10% [5–7]. Therefore, there is a need to develop image-based methods to enhance the throughput and accuracy of wheat head counting

in field conditions. This advancement will assist breeders in balancing the components of yield (plant number, head density, grains per head, grain weight) during breeding selections [8].

In recent years, with the advancement of computer vision and deep learning theories, methods based on deep neural network technology have made significant progress in addressing general object detection problems. Consequently, numerous researchers have endeavored to apply general object detection algorithms to the field of agricultural object detection. Xu et al. [9] employed the K-means clustering algorithm to automatically segment wheat head images and extract wheat head contour features, thereby significantly enhancing the efficiency and accuracy of wheat head counting. Wang et al. [10] utilized a multilevel neural network (SSRNET) for wheat head image segmentation, achieving rapid estimation of wheat head quantities under field conditions. Khaki et al. [11] drew inspiration from crowd counting research and constructed a fast and lightweight anchor-free wheat head counting network. Amirhossein et al. [12] applied automatic object enhancement technology to deep learning models and employed a hybrid AutoOLA-DL model to improve wheat head counting performance. Wang et al. [13] developed a lightweight apple detection model based on YOLOv5s, adopting a pruning and fine-tuning approach to maintain accuracy while ensuring overall model lightweightness for deployment on portable IoT devices. Sozzi et al. [14] extensively studied the performance differences of various YOLO networks in identifying white grape applications. He et al. [15] researched and simplified the YOLOv4 network structure, employing the K-means algorithm to recluster anchors, effectively addressing wheat head detection issues in natural scenes based on unmanned aerial vehicle platforms. Gong et al. [16] optimized the Spatial Pyramid Pooling (SPP) structure within the YOLOv4 network architecture to enhance feature learning capabilities, enlarging the convolutional network's receptive field, thereby improving recognition accuracy and speed. Zhao et al. [17] improved wheat head feature extraction by adding micro-scale detection layers and setting prior anchor boxes, optimizing the YOLOv5 network structure, significantly enhancing the detection accuracy of wheat head images captured by drones. Meng et al. [18] constructed an improved YOLOv7 model by incorporating attention modules, greatly enhancing wheat head detection effectiveness, and exploring the influence of attention module quantity and placement on the model. The primary research direction of the aforementioned studies can be summarized as augmenting various specially designed modules onto existing general object detectors according to the characteristics of the target objects to be detected, thereby achieving better detection performance.

However, existing wheat head detectors exhibit poor generalization in practical applications. This is attributed to several distinctive features commonly observed in wheat head detection scenarios: Different wheat varieties exhibit significant variations in wheat head color, size, and traits at various growth stages. Wheat fields are typically densely planted, leading to substantial mutual occlusion between wheat heads in images. Factors such as equipment, lighting, environmental conditions, and wind during image acquisition contribute to significant differences between wheat heads. Wheat is widely cultivated globally, and diverse wheat field environments present considerable variations based on climate, soil, altitude, etc. These characteristics result in substantial differences between wheat field scenes. Consequently, wheat head detectors trained on datasets specifically constructed for particular studies exhibit suboptimal performance when confronted with entirely new wheat field scenes beyond the scope of their training datasets [7].

To address the generalization issues of models, a larger dataset proves effective. In response to this challenge, the GWHD dataset was introduced by [8], and subsequently, the expanded GWHD2021 dataset was released [19]. The dataset encompasses 6515 wheat head images collected from 11 different countries and regions worldwide, making it the largest and most diverse publicly available wheat head dataset to date. However, compared to the commonly used MS-COCO dataset [20] in general object detection research, which comprises over 120,000 annotated images, the annotated images in GWHD2021 constitute

only 5% of MS-COCO's annotated images. This discrepancy arises from the challenging nature of creating wheat datasets due to the characteristics mentioned earlier in wheat head detection scenarios. Even with extensive training, annotators find the annotation process challenging and time-consuming [21]. This creates a paradox where despite the abundance of wheat head image data, the high cost in terms of both money and time required for annotation hinders their utilization. This underscores a major drawback of fully supervised object detection—overreliance on annotated datasets. In contrast, SSOD (semi-supervised object detection) proves effective in addressing the issue of insufficient annotated data.

SSL (Semi-Supervised Learning) is a learning paradigm that involves constructing models using both labeled and unlabeled data. These methods enhance learning performance by incorporating additional unlabeled instances, in contrast to supervised learning algorithms that rely solely on labeled data. SSL algorithms provide a means to extract latent patterns from unlabeled examples, reducing the dependency on a large number of labels [22]. DSSL (Deep Semi-Supervised Learning) is a specific research domain within SSL, exploring how to effectively leverage both labeled and unlabeled data through deep neural networks. Numerous DSSL methods have been proposed, categorizable into five main types based on distinctive features in semi-supervised loss functions and model designs. Generative methods: employing generative models to simulate the distribution of data and subsequently leveraging the generated data for learning purposes; a paradigmatic approach in this context is the utilization of Generative Adversarial Network (GAN). Consistency regularization methods: these methods encourage models to produce similar outputs for different views or perturbations of input data, thereby improving robustness to unlabeled samples. Graph-based methods: using graph structures to represent similarity relationships between samples, these methods leverage techniques like label propagation or semi-supervised graph convolutional networks for semi-supervised learning. Pseudo-labeling methods: these methods generate pseudo-labels for unlabeled samples by using the model's predictions, treating them as real labels for training. Hybrid methods: combining elements of the above four methods in a hybrid manner [23].

SSOD represents a research domain within DSSL, aiming to explore the application of semi-supervised methods to address object detection challenges. Traditionally, in the field of computer vision, semi-supervised methods have predominantly been employed for image classification tasks rather than object detection. The investigation into semi-supervised methods for object detection heavily relies on the foundational works of Sohn et al. [24] and Unbiased Teacher [25]. Sohn et al. [24] were pioneers in introducing pseudo-labeling methods to SSOD, establishing a paradigm where teacher and student models are concurrently trained. The teacher model utilizes pseudo-labels generated from unlabeled data to guide the training of the student model. Building upon this, Unbiased Teacher [25] introduced consistency regularization methods to SSOD. The unlabeled data fed into the teacher model undergoes mild data augmentation, while the unlabeled data for the student model undergoes strong data augmentation. Notably, the parameters of the teacher model are no longer frozen but are copied from the student model through the Exponential Moving Average (EMA) approach, a technique initially proposed by Tarvainen and Valpola [26] for solving classification problems. Subsequently, most SSOD methods adhere to the paradigms established by the aforementioned works, employing hybrid methods that combine pseudo-labeling and consistency regularization. Simultaneously training both teacher and student models, these methods use the teacher model's predictions on unlabeled data with confidence exceeding a threshold as pseudo-labels to guide the training of the student model. Our research aligns with these established paradigms.

Subsequent to these developments, the primary research focus within SSOD has been on optimizing the utilization of pseudo-labels and effectively distinguishing between reliable and unreliable pseudo-labels. Influential works in this direction include the following. Soft Teacher [27], which contends that pseudo-labels with confidence below a threshold contain valuable information and should not be discarded outright. Instead, they advocate for incorporating these pseudo-labels into the loss calculation after weighting. Humble

Teacher [28], employing an abundance of region proposals and soft pseudo-labels as training targets for the student model. Dense Teacher [29], a pioneering work implementing SSOD on a one-stage anchor-free detector, departing from the traditional two-stage anchor-based detectors used in prior SSOD research. Unbiased TeacherV2 [30], proposing a perspective shift in treating pseudo-label boxes not as a whole but as individual edges, placing trust only in edges with confidence surpassing a threshold. LabelMatch [31], advocating for varied confidence thresholds for pseudo-labels of different object categories rather than a uniform threshold. Efficient Teacher [32], introducing the first one-stage anchor-based semi-supervised object detector. Consistent Teacher [33], introducing three modules—Adaptive Anchor Assignment (ASA), 3D Feature Alignment Module (FAM-3D), and Gaussian Mixture Model (GMM)—to address the issue of pseudo-label oscillation during student model training. Our research draws inspiration from these aforementioned studies.

Several researchers have endeavored to employ SSOD methods to address challenges in agriculture, particularly in the realm of weed identification, a prevalent issue across various agricultural scenarios. The definition of "weed" varies depending on the cultivated crops, adding significant complexity to the annotation process. Kerdegari et al. [34] introduced a weed identification model using generative methods; opting not to rely on pseudo-labeling or consistency regularization methods, they trained a Generative Adversarial Network (GAN) on labeled data and utilized the generated weed images as part of their dataset. Jiang et al. [35] proposed a weed identification model based on graph-based methods, employing a Graph Convolutional Network (GCN) built on CNN features to construct a graph using extracted weed CNN features and their Euclidean distances. Shorewala et al. [36] presented a generalized semi-supervised weed identification model effective in both carrot and sugar beet datasets, requiring no fine-tuning. Menezes et al. [37] introduced a soybean field weed crop recognition system based on superpixels, detecting weeds in soybean fields as well as other crops beyond soybeans. Liu et al. [38] proposed a semi-supervised wheat field weed detector incorporating attention mechanisms to aid precise herbicide application. Benchallal et al. [39] introduced a weed identification system using only consistency regularization methods without employing any pseudo-labeling methods. Beyond weed identification, researchers have applied SSOD to crop recognition problems. Khaki et al. proposed DeepCorn [40], estimating corn kernel density in images of corn ears and predicting the number of kernels based on the estimated density map. Casado et al. [41] extended object detection to semi-supervised semantic segmentation, aiming to segment mature grapes in vineyards. Xu et al. [42], utilizing YOLOv5X, constructed a semi-supervised semantic segmentation model for Maize Seedling Leaf counting. Johanson et al. [43] introduced S3AD, a semi-supervised detection system based on contextual attention and selective tiling, addressing small apple detection. Across these studies, semi-supervised methods have demonstrated immense potential, effectively leveraging unlabeled data to enhance detector performance and successfully addressing the issue of insufficient annotations.

Fourati et al. [44] and Chen et al. (https://github.com/ksnxr/GWC_solution accessed on 14 October 2023) attempted to address the wheat head detection problem using semi-supervised methods. However, both approaches adopted ensemble learning, which combines predictions from multiple networks to harness the potential of multiple models. Nevertheless, this comes at the cost of significantly increased training time. In contrast to fully supervised methods, semi-supervised training itself incurs substantial time overhead due to handling unlabeled data and simultaneous training of both teacher and student models. If ensemble learning methods are additionally employed, the training time cost becomes intolerable. Therefore, we propose a novel wheat head detector, Wheat Teacher, which achieves convergence of metrics in less than a day when trained on the GWHD2021 dataset using a single RTX4090. Wheat Teacher employs the one-stage anchor-based detector, YOLOv5, as its backbone. Most semi-supervised object detection methods are implemented using either one-stage anchor-free detectors like FCOS [45] or two-stage anchor-based detectors like Faster R-CNN. However, one-stage anchor-based detectors

offer advantages such as high recall, numerical stability, and fast training speed, making them particularly suitable for applications with densely populated targets like wheat head detection. In the context of wheat head detection, Wheat Teacher incorporates our proposed innovative Pseudo-label Dynamic Allocator and Loss Dynamic Threshold. These components are specifically designed to enhance wheat head detection performance. Specifically, the main contributions of this study are as follows:

1.  Proposal of Wheat Teacher, a novel one-stage anchor-based semi-supervised wheat head detector employing YOLOv5 as its backbone. Achieving metrics comparable to various fully supervised object detectors utilizing 100% labeled data, Wheat Teacher demonstrates remarkable performance with only 20% labeled data.
2.  Proposal of an innovative Pseudo-label Dynamic Allocator for dynamically allocating pseudo-labels.
3.  Proposal of an innovative Loss Dynamic Threshold for adaptively filtering out irrelevant losses.

The remaining sections of this paper are organized as follows: Section 2 provides an introduction to the model and network architecture, offering detailed explanations of our semi-supervised training methodology. In Section 3, experimental details and results will be presented, comparing them with the results obtained through fully supervised methods. Subsequently, in Section 4, we discuss the methods employed in this study and propose future research directions. Finally, in Section 5, we conclude this paper.

## 2. Methods

### 2.1. Wheat Teacher

Wheat Teacher is a one-stage anchor-based semi-supervised wheat head detector that employs hybrid methods in the semi-supervised approach. The network architecture adopts a teacher-student paradigm, utilizing YOLOv5 as the backbone. Additionally, novel components, namely the Pseudo-label Dynamic Allocator and Loss Dynamic Threshold, are introduced specifically for wheat head detection scenarios to allocate pseudo-labels and filter losses.

The hybrid methods employed by Wheat Teacher combine pseudo-labeling methods and consistency regularization methods. These methods are based on two assumptions. The first assumption posits that using different data augmentations on the same image should yield consistent predictions from the same model. The second assumption suggests that if the predictions on the same image differ after weak and strong data augmentations, the predictions on the weakly augmented image should be considered more accurate, as it is easier to predict.

Building upon these assumptions, Wheat Teacher employs a teacher–student paradigm, consisting of two models: the teacher model and the student model. The basic training process involves weakly augmenting unlabeled data images and inputting them into the teacher model for predictions. The same images are then strongly augmented and input into the student model for predictions. The teacher model's predictions with confidence exceeding a threshold are treated as the ground truth for the unlabeled data images. These predictions are compared with the student model's predictions, guiding the calculation of the student model's loss. The predictions from the teacher model serve as pseudo-labels.

The training process of Wheat Teacher comprises several steps. Firstly, in the burn-in stage, the backbone model is pre-trained using labeled data. After pre-training, the model is duplicated into the teacher and student models. In the semi-supervised stage, unlabeled data undergo weak data augmentation (e.g., horizontal and vertical flips, cropping and resizing, rotations, mosaic) and are input into the teacher model to generate predictions. Pseudo-labels from the teacher model, exceeding a confidence threshold, are considered the ground truth for unlabeled data. After generating and filtering pseudo-labels, the unlabeled data previously input into the teacher model undergo strong data augmentation (e.g., mixup, random erasing). These augmented unlabeled data, along with pseudo-labels (treated as annotations for unlabeled data), are concatenated with weakly augmented la-

beled data and input into the student model. The student model simultaneously calculates losses for both unlabeled and labeled data, updating its parameters through backpropagation. After each iteration, the parameters of the teacher model are updated using an Exponential Moving Average (EMA) of the student model's parameters. This process repeats in subsequent iterations. Figure 1 illustrates the specific pipeline of Wheat Teacher.

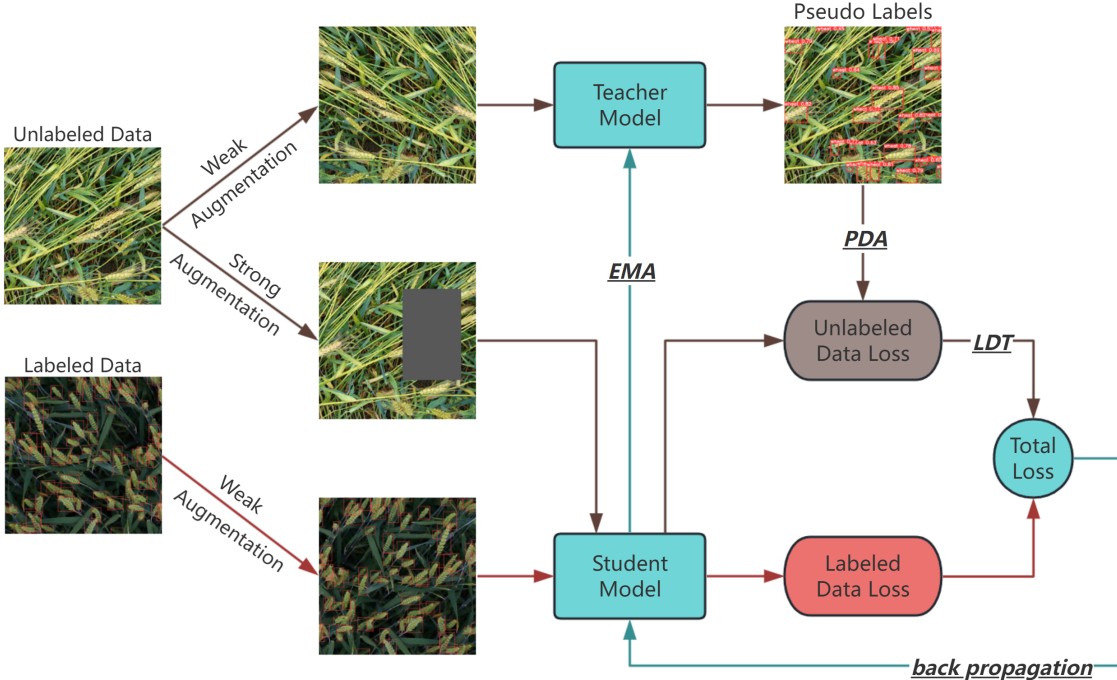

**Figure 1.** The pipeline of Wheat Teacher.

## 2.2. Pseudo-Label Dynamic Allocator (PDA)

The quality of predictions generated by the teacher model for unlabeled data is inconsistent, with most predictions being unsuitable as pseudo-labels for guiding the student model. Hence, there is a need for pseudo-label filtering. A common practice involves setting a pseudo-label threshold, categorizing pseudo-labels into reliable and unreliable binary distinctions. Early semi-supervised methods would outright discard unreliable pseudo-labels. However, contemporary semi-supervised approaches posit that even unreliable pseudo-labels may contain valuable information and should not be discarded outright; instead, they can be processed and utilized. For instance, allowing the loss computed from unreliable pseudo-labels to undergo weighted processing before backpropagation [27]. While setting a pseudo-label threshold allows for a quick and simple filtering of pseudo-labels, the overall scores of pseudo-labels tend to increase gradually throughout the entire semi-supervised training process. This phenomenon makes it challenging for the model to converge. We contend that the model's outputs consistently comprise both relatively correct and relatively incorrect predictions. Therefore, the threshold should dynamically adjust based on the overall distribution of pseudo-labels in each iteration. Additionally, we posit that a binary categorization of pseudo-labels is inappropriate. While unreliable pseudo-labels may contain useful information, there exists substantial variability among unreliable pseudo-labels. Some unreliable pseudo-labels contain minimal useful information, and attempting to leverage them may inevitably introduce noise, resulting in an unfavorable trade-off. Therefore, a more refined differentiation of unreliable pseudo-labels is warranted.

Therefore, we introduce the innovative Pseudo-label Dynamic Allocator, a module capable of autonomously adjusting thresholds based on the confidence distribution of pseudo-labels and categorizing them. The Pseudo-label Dynamic Allocator incorporates

two dynamic thresholds: the high-confidence threshold and the low-confidence threshold. The low-confidence threshold is set to 0.1 times the sum of confidences for all pseudo-labels generated in one iteration. Pseudo-labels with confidence below this threshold are considered unreliable and are directly discarded without participating in loss computation. After removing low-confidence pseudo-labels, the average confidence of all remaining pseudo-labels becomes the high-confidence threshold. Pseudo-labels with confidence above this threshold are considered reliable and directly contribute to loss computation. Pseudo-labels with confidence between the low- and high-confidence thresholds are considered uncertain. If an object's confidence in an uncertain pseudo-label exceeds 0.8, this pseudo-label participates in IoU loss computation. The specific process of the Pseudo-label Dynamic Allocator is outlined using pseudocode in Algorithm 1.

---

**Algorithm 1** Pseudo-label Dynamic Allocator

---

1: **for** $I = I_1, \ldots, I_N$ **do**            ▷ In each iteration, $I$ means one iteration.
2:      $\alpha = L_{obj} = L_{reg} = 0$
3:      $P^r = \varnothing$
4:      **for** $P = P_1, \ldots, P_N$ **do**            ▷ $P$ means pseudo-label.
5:          $\alpha = \alpha + P_{conf}$            ▷ $P_{conf}$ means confidence of pseudo-label.
6:      **end for**
7:      $\tau_l = \alpha * 0.1$            ▷ Calculate the low confidence threshold $\tau_l$.
8:      **for** $P = P_1, \ldots, P_N$ **do**
9:          **if** $P_{conf} \geqslant \tau_l$ **then**
10:             $P^r \cup \{P\}$            ▷ $P^r$ means reliable pseudo-labels.
11:          **end if**
12:      **end for**
13:      $\tau_h = \frac{1}{N} \sum_{n=1}^{N} P_{conf}^{r\,n}$            ▷ Calculate the high confidence threshold $\tau_h$.
14:      **for** $P^r = P_1^r, \ldots, P_N^r$ **do**
15:          **if** $P_{Conf}^r \geqslant \tau_h$ **then**
16:             $L_{obj} = L_{obj} + CE(P^r)$            ▷ $L_{obj}$ means object loss.
17:             $L_{reg} = L_{reg} + CIoU(P^r)$            ▷ $L_{reg}$ means regression loss.
18:          **else if** $P_{objconf}^r \geqslant 0.8$ **then**            ▷ $P_{objconf}^r$ means object confidence.
19:             $L_{reg} = L_{reg} + CIoU(P^r)$
20:          **end if**
21:      **end for**
22:      **return** $L_{obj}, L_{reg}$
23: **end for**

---

With the introduction of the Pseudo-label Dynamic Allocator, the loss formulation for Wheat Teacher is as follows:

$$L = L_s + \lambda L_u \tag{1}$$

In the formulation, where $L$ represents the total loss, $L_s$ is the loss computed from labeled data, $L_u$ is the loss computed from unlabeled data, and $\lambda$ is a weighted coefficient used to balance the losses between full-supervision and semi-supervision, we set it to 3. It is important to note that, since Wheat Teacher only detects the "wheat head" object, class loss is not computed.

The definition of $L_s$ is as follows:

$$L_s = \sum_{n=1}^{N} (CIoU(X_n^{reg}, Y_n^{reg}) + CE(X_n^{obj}, Y_n^{obj})) \tag{2}$$

$CIoU$ represents the Complete Intersection over Union loss function, $CE$ represents the cross-entropy loss function, $X_n$ represents the output of the student model, and $Y_n$ represents the ground truth.

The definition of $L_u$ is as follows:

$$L_u = L_u^{reg} + L_u^{obj} \tag{3}$$

$$L_u^{reg} = \sum_{n=1}^{N} (\mathbb{I}_{\left\{ p_n \geqslant \tau_h \text{ or } (p_n \geqslant \tau_l \text{ and } \hat{obj}_n > 0.8) \right\}} CIoU(X_n^{reg}, \hat{Y}_n^{reg})) \tag{4}$$

$$L_u^{obj} = \sum_{n=1}^{N} (\mathbb{I}_{\{ p_n \geqslant \tau_h \}} CE(X_n^{obj}, \hat{Y}_n^{obj})) \tag{5}$$

$\hat{Y}_n$ represents the pseudo-label, $p_n$ represents the overall confidence of this pseudo-label, $\hat{obj}_n$ represents the object confidence of this pseudo-label, $\tau_l$ represents the low-confidence threshold, $\tau_h$ represents the high-confidence threshold, $\mathbb{I}_{\{\cdot\}}$ represents an indicator function. If condition $\{\cdot\}$ is satisfied, the output is 1; otherwise, it is 0.

### 2.3. Loss Dynamic Threshold (LDT)

Even after filtering pseudo-labels through the Pseudo-label Dynamic Allocator, the remaining pseudo-labels are not entirely accurate because they are based on predictions from the teacher model, which is not flawless. Through experiments, we observed that in images treated as unlabeled data, for the same wheat head, the differences between the predictions of the student model, pseudo-labels, and ground truth are often minimal. However, the gap between the predictions of the student model and pseudo-labels can sometimes be larger than the gap between the predictions of the student model and the ground truth. We attribute this phenomenon to the fact that Wheat Teacher employs YOLOv5 as its backbone, which is a one-stage anchor-based detector. This type of detector outputs a large number of prediction boxes, necessitating post-processing to filter out predictions. YOLOv5 uses Non-Maximum Suppression (NMS) as its post-processing method, removing all candidate boxes except the optimal one. In the typical scenario of YOLOv5 forward inference, NMS has been empirically proven to be a highly effective post-processing technique. However, in the context of semi-supervised training, there is uncertainty regarding whether the bounding box selected by NMS as the optimal candidate, when compared to less optimal candidates, exhibits a minimal difference with the true ground truth.

The subtle variations in loss due to the uncertainty of pseudo-labels do not significantly benefit the performance of the student model. In fact, they may even mislead the student model, leading to unnecessary oscillations during the training process. Therefore, we innovatively propose the Loss Dynamic Threshold. In each iteration, it is crucial to determine which losses induced by pseudo-labels are necessary and which are unnecessary, in order to reduce unnecessary oscillations in the model training. We advocate for dynamically adjusting the Loss Dynamic Threshold based on the total loss value generated by pseudo-labels in each iteration. Subsequently, losses below this threshold should be discarded. However, if the model calculates the loss but does not immediately backpropagate and clear gradients after each loss computation, attempting to backpropagate all losses at once, the GPU must store the complete computation graph for each loss to ensure correct backpropagation. This results in a multiplicative increase in model memory usage. While setting the batch size for each iteration to 1 effectively addresses this issue, it leads to significant memory wastage, severely slowing down the model training speed.Therefore, to ensure training efficiency, we set the Loss Dynamic Threshold for each iteration to the peak value of the lowest 10% of losses generated by pseudo-labels in the previous iteration. Losses below the Loss Dynamic Threshold are considered unnecessary and are set to 0, while losses exceeding the threshold undergo normal backpropagation. The specific process of Loss Dynamic Threshold is outlined in Algorithm 2 using pseudocode.

---

**Algorithm 2** Loss Dynamic Threshold

---

1:  **for** $I = I_1, \ldots, I_N$ **do**                              ▷ In each iteration, $I$ means one iteration.
2:      **for** $l = l_1, \ldots, l_N$ **do**
3:          **if** $l \geqslant \tau$ **then**                              ▷ $\tau$ means dynamic loss threshold.
4:              $BackPropagation(l)$                              ▷ Backpropagate through the loss $l$.
5:              $L \cup \{l\}$                              ▷ $L$ is used to record all losses.
6:          **else**
7:              $L \cup \{l\}$
8:          **end if**
9:      **end for**
10:     **Sort**$(L)$                              ▷ Sort $L$ in **ascending** order.
11:     $\tau = L[\lceil 0.1 \times \mathbf{length}(L) \rceil]$                              ▷ Update $\tau$.
12:     $L = \varnothing$
13: **end for**

---

### 2.4. Backbone

The mainstream algorithms for general object detection can be broadly categorized into two types: two-stage detectors and one-stage detectors. Two-stage detectors, relying on region proposals, extract features through multi-stage networks, achieving higher accuracy at the cost of slower detection speed. Representative algorithms in this category include the R-FCN and R-CNN series (comprising R-CNN [46], Fast-RCNN [47], Faster-RCNN [48], Mask-RCNN [49]). On the other hand, one-stage detectors, based on region regression, directly extract features using a single-stage network, resulting in extremely fast detection. Notable algorithms in this category encompass SSD [50] and the YOLO series (encompassing YOLO [51], YOLO9000 [52], YOLOv3 [53], YOLOv4 [54], YOLOv5 (https://github.com/ultralytics/yolov5 accessed on 10 September 2023), YOLOX [55], YOLOv6 [56], YOLOv7 [57], etc.).

Given the application scenario of this research, Wheat Teacher opted for YOLOv5 as the backbone. YOLOv5 (You Only Look Once version 5) is a one-stage anchor-based detector, primarily comprising four components: input, backbone, neck, and head. The input component is responsible for image preprocessing, including adaptive anchor computation, various data augmentations, image data scaling, and other operations. YOLOv5's backbone is CSPDarknet53, which includes three main modules: CBS module, CSP module, and SPPF module. The core CBS module consists of three parts: two-dimensional convolution (Conv), batch normalization (BN), and Sigmoid-weighted Linear Unit activation function (SILU). The backbone extracts feature information from the image and combines it to form feature maps of different granularities. The neck section combines feature maps and extracts features through Upsample and Concatenate operations to enhance the robustness of the detection network. The head section outputs the target detection result, and the number of branches at the output varies depending on the detection scenario. Figure 2 illustrates the specific network structure of YOLOv5.

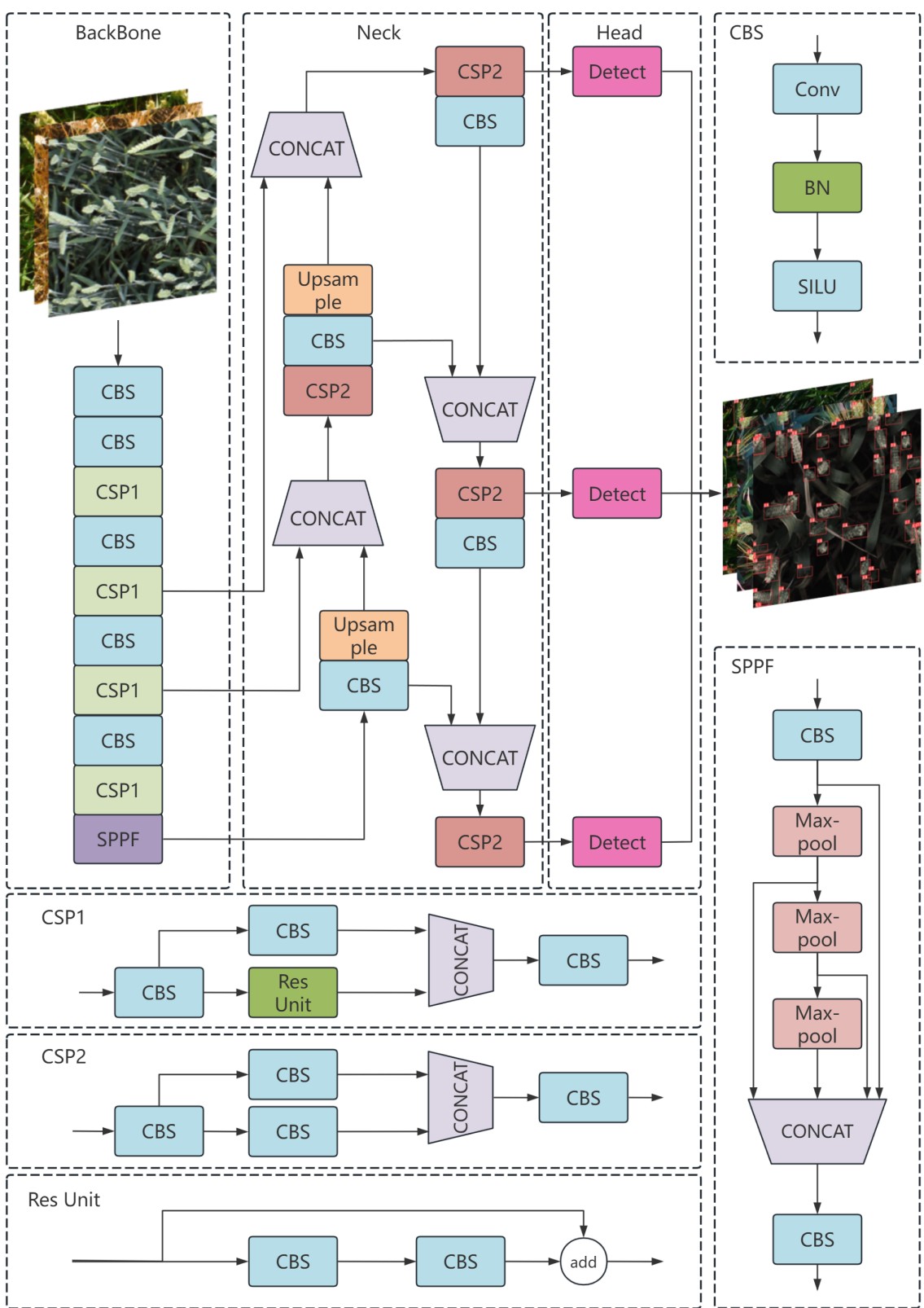

**Figure 2.** The network structure of YOLOv5.

## 3. Experimental Results

### 3.1. Dataset

In the specific domain of wheat head detection, research on existing technologies indicates limitations in proposed solutions, primarily addressing images from controlled

environments rather than those directly captured in the field [58]. Another limitation is that models in both the training and testing phases focus on the same type of wheat, leading to overfitting to that specific wheat type [7]. To overcome these limitations, this study decided to utilize the GWHD2021 dataset.

The Global Wheat Head Detection Dataset 2021 (GWHD2021 [19]) (http://www.global-wheat.com/ accessed on 6 October 2023) is a large-scale wheat head object detection dataset based on optical images. It is currently the largest publicly available Wheat Head dataset. The Global Wheat Head Detection Competition was initiated by the Global Wheat Dataset consortium in 2020, providing the public dataset Global Wheat Head Detection Dataset 2020 (GWHD2020) for the competition. In 2021, the Global Wheat Dataset consortium launched the Global Wheat Challenge 2021 and expanded the GWHD2020 dataset in terms of wheat head diversity, label reliability, and data size, releasing GWHD2021.

GWHD2021 comprises 6515 RGB wheat head images collected from 11 different countries and regions worldwide, including Europe, North America, Asia, and Australia. Each image has a resolution of 1024 × 1024 pixels, and the labels include a total of over 270,000 wheat heads. The row spacing in the wheat fields varies from 12.5 cm to 30.5 cm. In addition to variations in row spacing, each field has different planting densities. Furthermore, the soil characteristics in the growth areas vary from mountainous regions to traditional irrigated farmlands, leading to differences in color and lighting conditions. These images were captured using various cameras at different distances from wheat heads (ranging from 1.8 m to 3 m). All these factors ensure the diversity of wheat heads in the GWHD2021 dataset. Some sample images from GWHD2021 are shown in Figure 3.

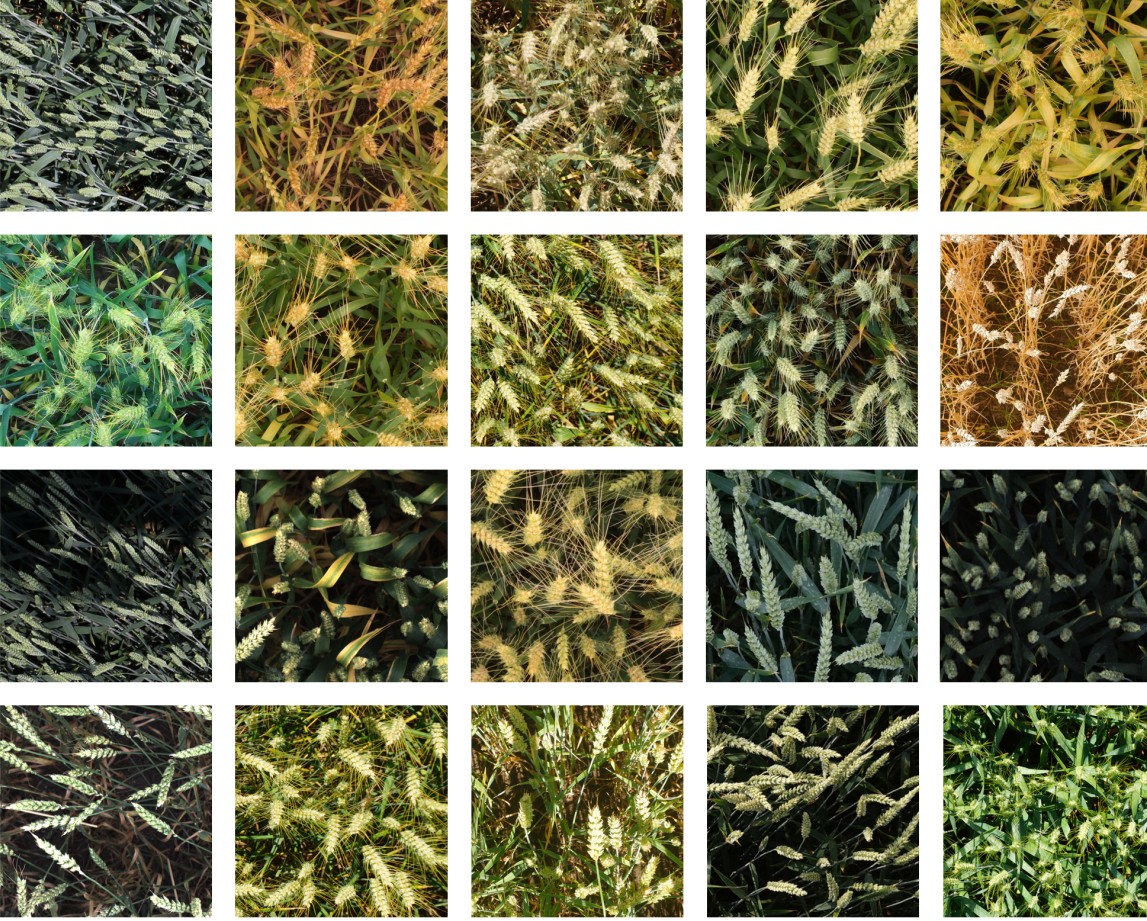

**Figure 3.** Some example images in the GWHD2021.

The semi-supervised training method requires a significant amount of images as unlabeled data. To ensure a fair comparison with other methods, this study did not introduce additional images beyond the GWHD2021 dataset for extra training. Instead, a total of 6000 images were randomly selected from the GWHD2021 dataset, which originally comprises 6515 images, to form a new training set. The remaining 515 images were used as a new validation set for reporting performance.

### 3.2. Evaluation Metric

To ensure the fairness and comparability of experimental results, widely used evaluation metrics in existing object detection methods were employed to assess the detection performance of the proposed model. These metrics include accuracy, recall, F1 score, and mean average precision (mAP). Specifically, accuracy measures the precision of the algorithm, while recall gauges the completeness of the image recognition results. The F1 score serves as a comprehensive evaluation metric for the model's detection accuracy, being the harmonic mean of precision and recall.

$$Precision = \frac{TP}{TP + FP} \tag{6}$$

$$Recall = \frac{TP}{TP + FN} \tag{7}$$

$$F1 = \frac{2 \cdot Precision \cdot Recall}{Precision + Recall} \tag{8}$$

In the above formulas, true positives (TP) represent the number of correctly classified wheat heads, where both the detection result and ground truth are wheat heads. False negatives (FN) indicate the number of incorrectly classified wheat heads, where the detection result is the background, and the ground truth is a wheat head. False positives (FP) denote the number of incorrectly classified backgrounds, where the detection result is a wheat head, and the ground truth is the background. True negatives (TN) signify the number of correctly classified backgrounds, where both the detection result and ground truth are backgrounds.

$$AP_n = \int_0^1 P(R)dR \tag{9}$$

$$mAP = \frac{1}{N} \sum_{n=1}^{N} AP_i \tag{10}$$

mAP (Mean Average Precision) is a crucial evaluation metric for measuring the performance of object detectors. It is defined as the area under the precision–recall curve, averaged across all classes, providing a measure of the overall performance of the model. mAP@0.5 assesses the model's performance on partially overlapping objects at an IoU (Intersection over Union) threshold of 0.5. mAP@0.5:0.95 further considers IoU thresholds ranging from 0.5 to 0.95, comprehensively evaluating the model's performance across various degrees of target overlap. As wheat head is the sole detection target in this study, the mAP value is equivalent to the AP value in this context.

### 3.3. Hyperparameter Settings

Wheat Teacher employs YOLOv5L 7.0 version as its backbone, with an input image size set to 1024 and a batch size of 6. The learning rate remains fixed at 0.01 throughout the training process. The rate for the exponential moving average (EMA) is set to 0.999. The training is conducted for a total of 350 epochs, comprising a Burn-in stage of 100 epochs during full-supervision training and a subsequent Semi-supervised stage of 150 epochs. In this study, for fair performance comparison, no transfer learning is applied, and no pre-trained weights are utilized. All models are initialized randomly.

### 3.4. Experimental Results Using Different Proportions of Labeled Data

In the experiment, we randomly partition the train set images into labeled and unlabeled data at different ratios (1% and 99%, 2% and 98%, 5% and 95%, 10% and 90%, 20% and 80%). The labels of unlabeled data are not used in the experiment. In a single experiment, Wheat Teacher first enters the burn-in stage, conducting fully supervised training using labeled data, and reports performance on the validation set after training. Subsequently, Wheat Teacher enters the semi-supervised stage, conducting training using both labeled and unlabeled data, and reports performance again on the validation set after training. The experimental results are shown in Table 1.

**Table 1.** Experimental results of labeled data with different proportions.

| Labeled Data | Unlabeled Data | mAP@0.5 | mAP@0.5:0.95 | Precision | Recall | F1-Score |
|---|---|---|---|---|---|---|
| 1% | — | 14.5 | 5.1 | 22.1 | 24.0 | 23.0 |
| 1% | 99% | 40.3 | 17.6 | 57.0 | 38.2 | 45.7 |
| 2% | — | 33.1 | 18.3 | 43.1 | 41.8 | 42.4 |
| 2% | 98% | 70.8 | 36.6 | 72.1 | 64.5 | 68.1 |
| 5% | — | 78.6 | 38.9 | 81.3 | 70.9 | 75.7 |
| 5% | 95% | 83.7 | 45.5 | 84.3 | 78.9 | 81.5 |
| 10% | — | 87.2 | 46.2 | 87.6 | 79.4 | 83.2 |
| 10% | 90% | 90.8 | 51.2 | 90.9 | 83.3 | 86.9 |
| 20% | — | 90.7 | 49.3 | 90.9 | 84.3 | 87.4 |
| 20% | 80% | 92.8 | 53.1 | 92.5 | 86.9 | 89.6 |

The experimental results indicate that by employing semi-supervised training with unlabeled data, Wheat Teacher achieved a significant improvement in metrics compared to the fully supervised training stage at any data ratio. Specifically, when using only 1% labeled data, Wheat Teacher achieved a 25.8% improvement in mAP0.5, a 12.5% improvement in mAP0.5:0.95, a 34.9% improvement in Precision, a 14.2% improvement in Recall, and a 22.7% improvement in F1-Score. When using only 2% labeled data, Wheat Teacher achieved a 37.7% improvement in mAP0.5, a 18.3% improvement in mAP0.5:0.95, a 29% improvement in Precision, a 22.7% improvement in Recall, and an 25.7% improvement in F1-Score. With only 5% labeled data, Wheat Teacher obtained a 5.1% improvement in mAP0.5, a 6.6% improvement in mAP0.5:0.95, a 3% improvement in Precision, a 8% improvement in Recall, and a 5.8% improvement in F1-Score. Using 10% labeled data, Wheat Teacher achieved a 3.6% improvement in mAP0.5, a 5% improvement in mAP0.5:0.95, a 3.3% improvement in Precision, a 3.9% improvement in Recall, and a 3.7% improvement in F1-Score. With only 20% labeled data, Wheat Teacher obtained a 2.1% improvement in mAP0.5, a 3.8% improvement in mAP0.5:0.95, a 1.6% improvement in Precision, a 2.6% improvement in Recall, and a 2.2% improvement in F1-Score.

Figures 4–6 depict the variations of Loss, mAP@0.5, and mAP@0.5:0.95 throughout the experimental training process, respectively. It can be observed that after concluding the fully supervised training phase, at the commencement of the semi-supervised training phase, the model's accuracy undergoes significant oscillations. However, during the subsequent training, the accuracy rapidly rebounds, eventually surpassing the results of fully supervised training. The oscillations observed after entering the semi-supervised training phase are one of the characteristics of semi-supervised training methods. However, in the general object detection domain, the amplitude of these oscillations is generally limited and should not be as extensive as depicted in the above figures. We attribute this phenomenon to the specific application scenario of this research, characterized by a single, highly dense object class. Eliminating such oscillations will be a future research direction.

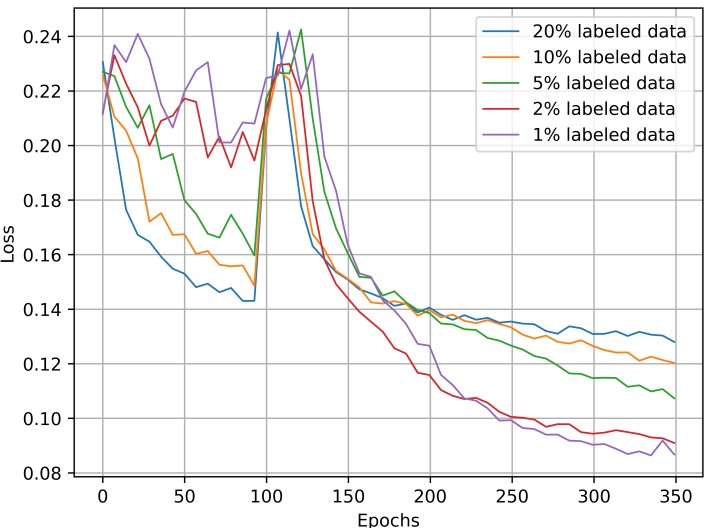

**Figure 4.** Graphs depicting loss curves.

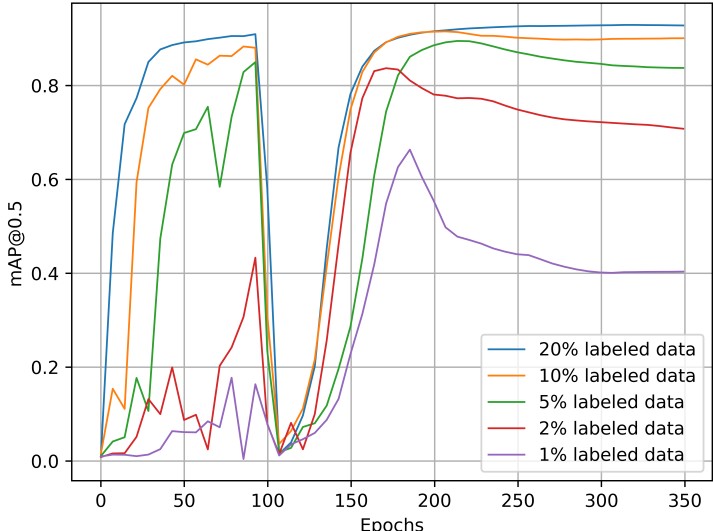

**Figure 5.** Graphs depicting mAP0.5 curves.

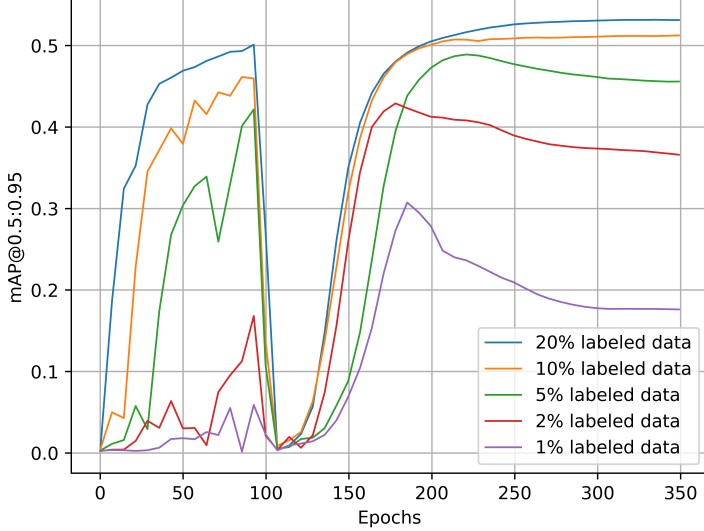

**Figure 6.** Graphs depicting mAP0.5:0.95 curves.

Figures 7 and 8 displays the precision–recall (PR) curve and the F1-Score curve after the completion of training. From the results above, it is evident that as the proportion of labeled data used in the fully supervised training phase increases, there is a noticeable improvement in the model's accuracy. This indicates that the model has not overfit, and the semi-supervised approach indeed has the potential to further enhance the model's accuracy. Semi-supervised methods show great promise in the field of wheat head detection.

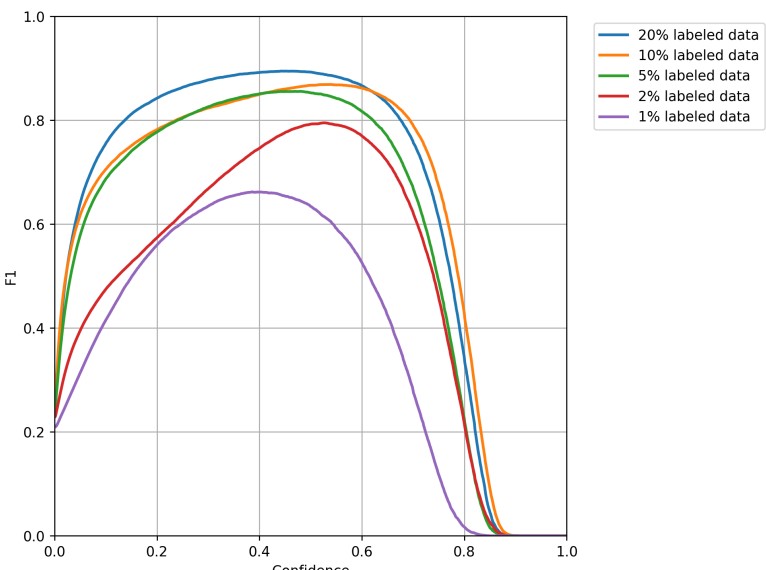

**Figure 7.** Graphs depicting F1 curves.

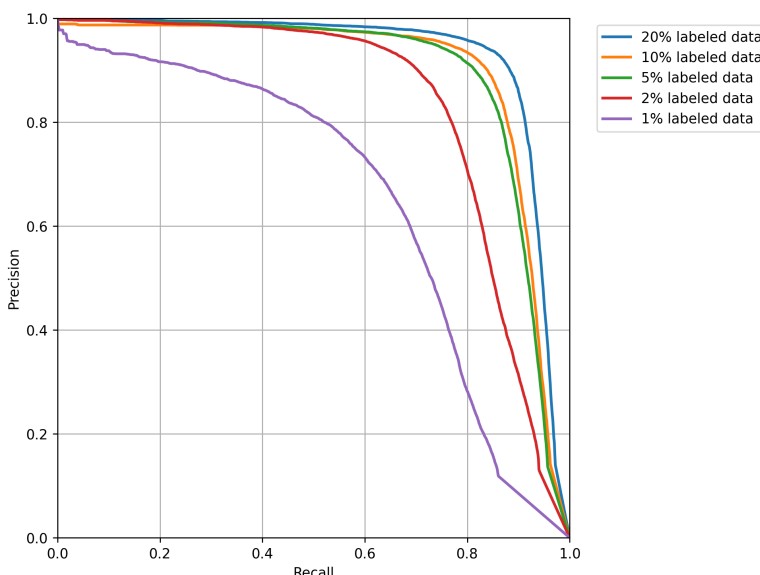

**Figure 8.** Graphs depicting PR curves.

### 3.5. Experimental Results Compared with Other Fully Supervised Object Detectors

The semi-supervised training method requires a substantial amount of images as unlabeled data. When compared with other fully supervised methods, we randomly divided the train set into labeled data and unlabeled data at a ratio of 20% to 80%. The labels of unlabeled data are not utilized by Wheat Teacher during the experiments. The experimental results are presented in Table 2.

We compared the predictions of Wheat Teacher with four fully supervised object detection models: Faster-RCNN, YOLOv5, YOLOX, and YOLOv7. The results show that

with only 20% of the training set labels, Wheat Teacher achieved highly competitive results. Wheat Teacher's metrics surpassed YOLOX, significantly outperformed Faster-RCNN, and the differences with YOLOv5 and YOLOv7 were within 1%, demonstrating the potential of semi-supervised methods in addressing wheat head detection. Figure 9 provides examples of prediction differences between Wheat Teacher, YOLOv5, and YOLOv7.

**Table 2.** Experimental results compared with other fully supervised object detectors on GWHD2021.

| Type | Method | mAP@0.5 | Precision | Recall | F1-Score |
|------|--------|---------|-----------|--------|----------|
| Two-stage Anchor-based | Faster-RCNN | 79.5 | 77.5 | 83.1 | 80.2 |
| One-stage Anchor-based | YOLOv7 | 93.6 | 92.5 | 87.4 | 89.8 |
| One-stage Anchor-free | YOLOX | 90.6 | 91.7 | 85.2 | 88.3 |
| Backbone | YOLOv5 | 93.1 | 92.2 | 87.1 | 89.5 |
| Ours | Wheat Teacher | 92.9 | 92.4 | 86.8 | 89.5 |

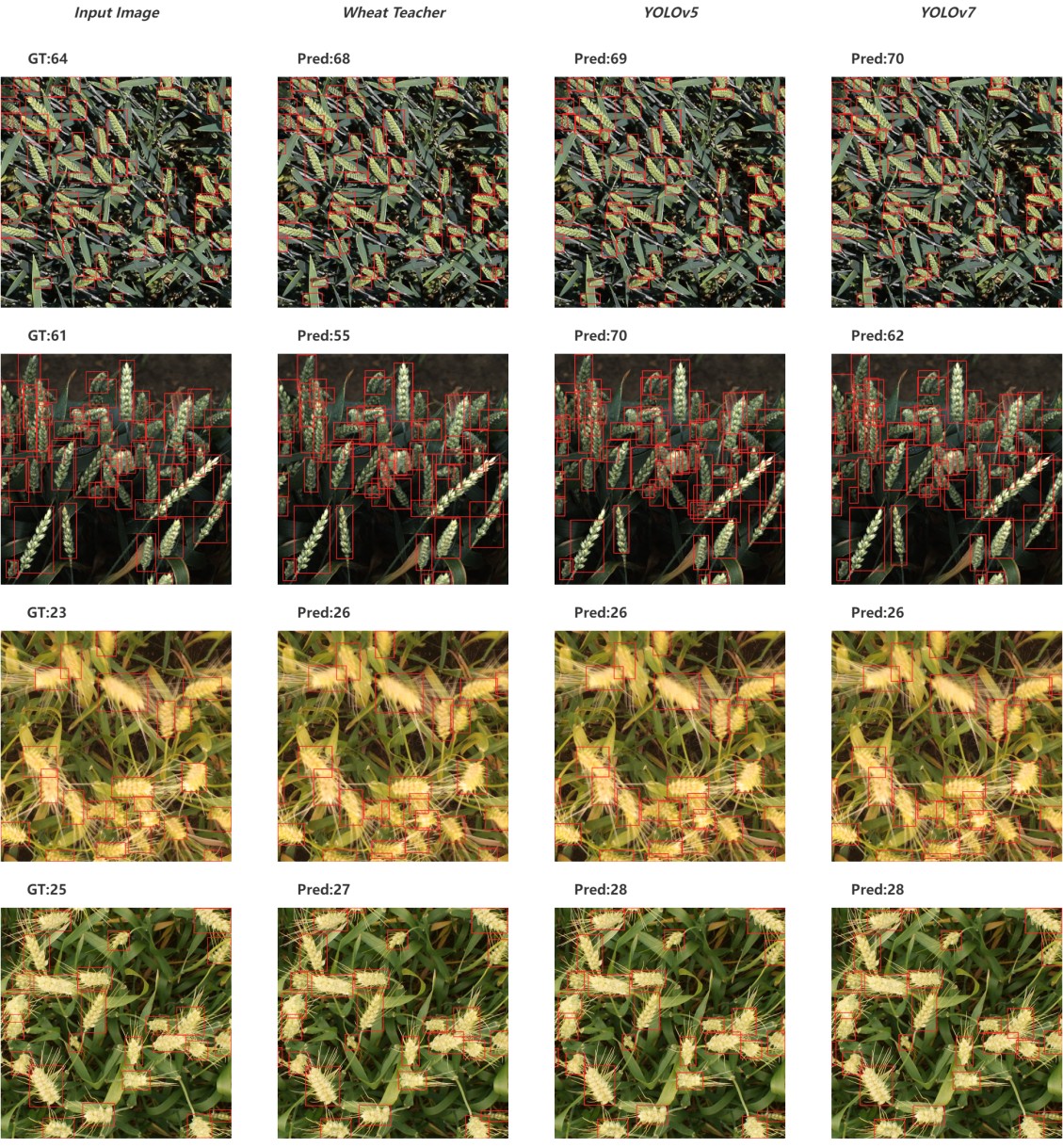

**Figure 9.** Differences in predictions under four different situations. The red box represents the wheat head. GT represents the number of wheat heads ground truth, and Pred represents the number of wheat heads predicted by the model.

### 3.6. Experimental Results Compared with Other Semi-Supervised Object Detectors

We conducted comparative experiments between Wheat Teacher and two existing semi-supervised object detection methods, Unbiased Teacher [25] and Efficient Teacher [32], at GWHD2021. The dataset was randomly sampled to consist of 20% labeled data and 80% unlabeled data, with all models utilizing the same random sampling scheme. The experimental results are presented in Table 3.

**Table 3.** Experimental results compared with other semi-supervised object detectors on GWHD2021.

| Method | mAP@0.5 | Precision | Recall | F1-Score | Training Time |
|---|---|---|---|---|---|
| Unbiased Teacher [25] | 90.8 | 90.5 | 83.9 | 87.1 | 29 h |
| Efficient Teacher [32] | 91.1 | 91.1 | 85.9 | 88.4 | 20 h |
| Wheat Teacher | 92.9 | 92.4 | 86.8 | 89.5 | 22 h |

From the experimental results, it can be observed that our method outperforms both Unbiased Teacher and Efficient Teacher across all metrics, except for a slightly longer training time compared to Efficient Teacher. This improvement may be attributed to the additional processing of the student network's predictions in our method, such as filtering losses, compared to Efficient Teacher.

### 3.7. Ablation Experiment

To validate the impact of our proposed Pseudo-label Dynamic Allocator and Loss Dynamic Threshold, we conducted a series of ablation experiments. We first removed the Loss Dynamic Threshold from Wheat Teacher and replaced the Pseudo-label Dynamic Allocator with a static threshold (set to 0.7), then reported the performance. Next, we replaced the static threshold with the Pseudo-label Dynamic Allocator and reported the performance. Finally, we introduced the Loss Dynamic Threshold into the network and reported the performance. All ablation experiments were conducted using 10% labeled data and 90% unlabeled data for training, and performance was evaluated on the validation set. The experimental results are presented in Table 4.

**Table 4.** Ablation experiment results.

| Pseudo-Label Threshold | Loss Threshold | mAP@0.5 |
|---|---|---|
| Static-Threshold (0.7) | × | 88.3 |
| PDA | × | 89.9 |
| PDA | LDT | 90.8 |

The results indicate that by introducing the Pseudo-label Dynamic Allocator, Wheat Teacher's mAP0.5 improved by 1.6%. Additionally, with the introduction of the Loss Dynamic Threshold, Wheat Teacher's mAP0.5 further increased by 0.9%, confirming the effectiveness of both the Pseudo-label Dynamic Allocator and the Loss Dynamic Threshold.

## 4. Discussion

Wheat Teacher is implemented based on a one-stage anchor-based detector, whereas most semi-supervised object detection methods are implemented on one-stage anchor-free detectors and two-stage anchor-based detectors. The reason for not using one-stage anchor-based detectors in many semi-supervised object detection methods lies in the fact that, compared to anchor-free detectors with no anchor design and two-stage detectors with multi-level filtering, one-stage anchor-based detectors tend to output more densely packed prediction boxes. This leads to a severe imbalance between positive and negative samples during semi-supervised training, and the issue of imbalance becomes more pronounced in detection scenarios with denser targets, such as wheat fields. To address this imbalance, we propose the Pseudo-label Dynamic Allocator and Loss Dynamic Threshold.

Before the one-stage anchor-based detector outputs the final predictions, it undergoes non-maximum suppression (NMS) to remove most irrelevant anchors. In essence, NMS can be considered as a filtering process on pseudo-labels based on a threshold, where only bounding boxes passing through the filter are deemed trustworthy pseudo-labels. Similar to NMS, which compares bounding boxes with each other, we believe that, instead of using a fixed static threshold set by hyperparameters, a more effective threshold should be determined by comparing the differences between bounding boxes. We propose the Pseudo-label Dynamic Allocator, which filters pseudo-labels based on the relationship between the confidence of pseudo-labels and the sum of pseudo-label confidences. Specifically, we set 0.1 times the sum of pseudo-label confidences as the low-confidence threshold and the mean of pseudo-label confidences as the high-confidence threshold.

Even after filtering, the remaining pseudo-labels inevitably contain errors. In dense target prediction scenarios, the model generates numerous bounding boxes. These bounding boxes, influenced by each other during non-maximum suppression, might replace more accurate bounding boxes due to subtle differences in confidence. While such differences might be inconsequential in other application scenarios, in semi-supervised training, these subtle differences can accumulate and mislead the student model throughout the training process. The objective of semi-supervised training is to enable the student model to learn correct features from unlabeled data. If the learned features are very small or even incorrect, we should abandon learning these features during training. Therefore, we propose the Loss Dynamic Threshold, which considers the peak in the lowest 10% of losses for each iteration as the threshold. This threshold is then used to filter out losses lower than the threshold in the next iteration. The effectiveness of Wheat Teacher in wheat head detection is adequately demonstrated in the experiments, and the roles of Pseudo-label Dynamic Allocator and Loss Dynamic Threshold are extensively validated in the ablation studies.

During the experimental process, we observed limitations when applying semi-supervised training methods to wheat head detection. If the labeled data is insufficient, it results in the initial model trained in the fully supervised training phase being suboptimal. Consequently, even when transitioning to semi-supervised training, the model's performance ceiling remains low. However, when an ample amount of labeled data is used in the fully supervised training phase, semi-supervised training methods demonstrate their potential. In comparison to other fully supervised object detection models that converge rapidly, semi-supervised training methods can leverage a large quantity of unlabeled data to continuously enhance their detection performance, exhibiting a higher performance ceiling than the backbone alone. Addressing the challenge of training a superior initial model with fewer labeled data points is expected to be a crucial avenue for future research.

## 5. Conclusions

This article delves into the significance and challenges of wheat head object detection. It introduces semi-supervised object detection methods and their applications in agriculture, subsequently proposing a one-stage anchor-based semi-supervised wheat head detector named "Wheat Teacher." Wheat Teacher amalgamates two semi-supervised methods, pseudo-labeling and consistency regularization, and integrates two novel dynamic threshold components, namely the Pseudo-label Dynamic Allocator and Loss Dynamic Threshold. We conducted detailed experiments on the largest public wheat head dataset, GWHD2021. Compared with various types of detectors, Wheat Teacher achieved an mAP0.5 of 92.8% with only 20% labeled data. This result surpassed the test outcomes of two fully supervised object detection models trained with 100% labeled data, and the difference with the other two fully supervised models trained with 100% labeled data was within 1%. Furthermore, Wheat Teacher shows improvements of 2.1%, 3.6%, 5.1%, 37.7%, and 25.8% in mAP0.5 across different labeled data usage ratios of 20%, 10%, 5%, 2%, and 1%, respectively. These experiments validate the effectiveness of our semi-supervised approach, highlighting the significant potential of Wheat Teacher in wheat head detection.

**Author Contributions:** Conceptualization, R.Z. and Y.S.; methodology, R.Z.; software, R.Z.; validation, R.Z.; formal analysis, R.Z.; investigation, R.Z.; resources, R.Z.; data curation, R.Z.; writing—original draft preparation, R.Z.; visualization, R.Z.; writing—review and editing, Y.S., M.Y., Z.Q., L.Z. and W.L.; supervision, Y.S.; project administration, Y.S.; funding acquisition, Y.S. All authors have read and agreed to the published version of the manuscript.

**Funding:** This research was funded by National Natural Science Foundation of China under Grant 61972147.

**Institutional Review Board Statement:** Not applicable.

**Data Availability Statement:** The original contributions presented in the study are included in the article, further inquiries can be directed to the corresponding author.

**Conflicts of Interest:** The authors declare no conflicts of interest.

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
