# Peer review of "Wheat Teacher: A One-Stage Anchor-Based Semi-Supervised Wheat Head Detector Utilizing Pseudo-Labeling and Consistency Regularization Methods"

_agriculture, doi:10.3390/agriculture14020327_

Round 1

Reviewer 1 Report

Comments and Suggestions for Authors

This article introduces a one-stage anchor-based semi-supervised wheat head detector, named "Wheat Teacher," which combines two semi-supervised methods, pseudo-labeling and consistency regularization. Additionally, two novel dynamic threshold components, Pseudo-label Dynamic Allocator and Loss Dynamic Threshold, were designed. In my opinion, it is a good article, written in a clear and straightforward manner. The problem addressed in the article is well introduced, along with its resolution.

To highlight something from the article, I think it would be interesting to conduct a time comparison. We are aware that conducting such tests requires significant resources and time, so a way to complement the article would be with this small comparison.

Author Response

We sincerely appreciate your professional review of our article.
Following your valuable suggestions, we have added some additional experiments to compare our method with several existing semi-supervised approaches, while also comparing the runtime.
Moreover, based on your and other reviewers' valuable feedback, we have extensively revised the previous manuscript. Please refer to the cover letter and the latest version of the manuscript for specific changes. We have listed all modifications in the cover letter and highlighted them in the latest manuscript.

Reviewer 2 Report

Comments and Suggestions for Authors

This work describes a teacher-student approach to wheat counting. Here are my considerations.

1.       The Introduction can provide further highlights on recent advancements in object detection in the agricultural field, as it is too focused on the teacher-student approach. I suggest broadening the scope of the state-of-the-art, for example, by adding the following relevant citations:

a.  https://doi.org/10.1016/j.biosystemseng.2021.08.015

b.  https://doi.org/10.3390/agronomy12020319

2.       The authors must further stress the real proposal of this work, which is the loss.

3.       How did the authors select the thresholds for the loss? Why did they use the provided hyperparameters? Please provide further arguments.

4.       The comparison is performed against fully supervised object detectors. However, similar methods should also be compared (e.g., Soft Teacher or similar). Please extend the experimental section.

5.       Check the formatting (e.g., specify the journal).

For all these reasons, I suggest the manuscript be considered after a major revision.

Author Response

We sincerely appreciate your professional review of our article, and here are our responses to your suggestions:
1.The Introduction can provide further highlights on recent advancements in object detection in the agricultural field, as it is too focused on the teacher-student approach. I suggest broadening the scope of the state-of-the-art, for example, by adding the following relevant citations:
a.https://doi.org/10.1016/j.biosystemseng.2021.08.015
b.https://doi.org/10.3390/agronomy12020319
We have significantly expanded the second paragraph of the Introduction, adding an overview of the latest advancements in object detection in the agricultural domain, including the literature you suggested.
2.The authors must further stress the real proposal of this work, which is the loss.
If we consider the predictions of the student model, pseudo-labels, and ground truth as points in the same coordinate system, the loss computed by the model can be viewed as a vector pointing from one point to another. Assuming the prediction of the student model is point S, the pseudo-label is point P, and the ground truth is point G, then the training process of the model can be seen as our desire for the vector starting from point S to point towards point G. However, we do not know the specific coordinates of point G; we only know that point P may be close to point G, and the exact coordinates of point P. Therefore, we can only calculate a vector pointing from point S to point P, hoping that advancing along this vector will bring point S closer to point G. When the Euclidean distance between points P, U, and G is small (i.e., when the computed loss is small), advancing along the vector from point S to point P may not bring point S closer to point G, and may even move it further away from point G. However, when the Euclidean distance between point S and points U, G is large (i.e., when the computed loss is large), advancing along the vector from point S to point P may bring point S closer to point G. The distance that point S advances is determined by the vector-weighted learning rate (i.e., the loss multiplied by the learning rate). As long as we use a stable low learning rate, we can be sure that the features learned by the model from the pseudo-labels (computed loss) will always move the model in the correct direction. Therefore, we believe that the losses should be filtered based on the magnitude of the loss generated by the pseudo-labels.
3.How did the authors select the thresholds for the loss? Why did they use the provided hyperparameters? Please provide further arguments.
We set the Loss Dynamic Threshold for each iteration to the peak value of the lowest 10% of losses generated by pseudo-labels in the previous iteration. Specifically, within an iteration, we record all the generated losses. Then, after an iteration ends, we sort all the recorded losses in ascending order. Finally, from the sorted losses, we select the peak value of the lowest 10% of losses as the next iteration's Loss Dynamic Threshold. We adopt this approach because experimental results indicate that such threshold selection helps to exclude a minority of excessively small losses in each iteration, leading to improved performance metrics. However, it's worth noting that since the number of losses generated in each iteration varies based on the batch size, the threshold selection method used in this study is strongly correlated with the GPU memory of the training device. All experiments in this study were conducted using a 4090 GPU, as mentioned in the paper. If other devices are used, further adjustment of hyperparameters may be necessary, which will be left for future research.
4.The comparison is performed against fully supervised object detectors. However, similar methods should also be compared (e.g., Soft Teacher or similar). Please extend the experimental section.
We have expanded the experimental section to include comparisons with two existing semi-supervised object detection methods, namely Unbiased Teacher and Efficient Teacher.
5.Check the formatting (e.g., specify the journal).
We have thoroughly reviewed and corrected all our references.
Additionally, based on your and other reviewers' valuable feedback, we have extensively revised the previous manuscript. Please refer to the cover letter and the latest version of the manuscript for specific changes. We have listed all modifications in the cover letter and highlighted them in the latest manuscript.

Reviewer 3 Report

Comments and Suggestions for Authors

The manuscript introduces a wheat head detection module called "Wheat Teacher," leveraging deep learning techniques. This module integrates two semi-supervised approaches, namely pseudo-labeling and consistency regularization, to enhance the precision of wheat spike identification—a concept of notable interest. Nevertheless, the current version of the manuscript reveals certain weaknesses. Adequate revisions to the following points should be undertaken to justify the recommendation for publication.

1.      Why was the YOLOv5 model chosen as the backbone network in the Wheat Head Detector proposed by the authors for detecting wheat spikes? Additionally, what considerations led them to prefer YOLOv5 over newer versions such as YOLOv6, YOLOv7, and YOLOv8 for this specific task?

2.      It is better to relocating the information presented in lines 178-183, i.e, "We conducted extensive experiments on …….. with 100% labeled data was within 1%". This information would be more suitably positioned in the results section rather than the introduction.

3.      In Figure 4, please ensure that distinct captions are assigned to each graph using labels (a), (b), and (c). Additionally, the visibility of these graphs is suboptimal, it is better to enlarge the size of the plots to enhance clarity and improve overall visibility.

4.       In Figure 5, distinct captions for each curve should be presented using labels (a) and (b). Additionally, enhance the font size of the text and data within these curves to improve visibility.

Author Response

We sincerely appreciate your professional review of our article, and here are our responses to your suggestions:
1.Why was the YOLOv5 model chosen as the backbone network in the Wheat Head Detector proposed by the authors for detecting wheat spikes? Additionally, what considerations led them to prefer YOLOv5 over newer versions such as YOLOv6, YOLOv7, and YOLOv8 for this specific task?
In the field of semi-supervised learning, researchers have traditionally focused more on novel semi-supervised training methods rather than on the choice of backbone. The improvement in metrics brought by a better backbone may not accurately reflect the effectiveness of the proposed semi-supervised training methods. Therefore, to ensure a fair comparison among various semi-supervised methods, researchers commonly adopt the same backbone, such as FasterRCNN, FCOS, and YOLOv5. We followed this paradigm in our study.
2.It is better to relocating the information presented in lines 178-183, i.e, "We conducted extensive experiments on …….. with 100% labeled data was within 1%". This information would be more suitably positioned in the results section rather than the introduction.
We have relocated this information to the results section as suggested.
3.In Figure 4, please ensure that distinct captions are assigned to each graph using labels (a), (b), and (c). Additionally, the visibility of these graphs is suboptimal, it is better to enlarge the size of the plots to enhance clarity and improve overall visibility.
We have split and enlarged Figure 4 for better clarity and readability.
4.In Figure 5, distinct captions for each curve should be presented using labels (a) and (b). Additionally, enhance the font size of the text and data within these curves to improve visibility.
Similar adjustments have been made to Figure 5 as in Figure 4.
Furthermore, based on your and other reviewers' valuable feedback, we have extensively revised the previous manuscript. Please refer to the cover letter and the latest version of the manuscript for specific changes. We have listed all modifications in the cover letter and highlighted them in the latest manuscript.

Round 2

Reviewer 2 Report

Comments and Suggestions for Authors

The paper can be considered for publication.

Reviewer 3 Report

Comments and Suggestions for Authors

All the comments have been considered in this version.  So this manuscript can be accepted for publication.